# Degradation of Polypropylene and Polypropylene Compounds on Co-Rotating Twin-Screw Extruders

**DOI:** 10.3390/polym17111509

**Published:** 2025-05-28

**Authors:** Paul Albrecht, Matthias Altepeter, Florian Brüning

**Affiliations:** Kunststofftechnik Paderborn, Paderborn University, 33098 Paderborn, Germanyflorian.bruening@ktp.uni-paderborn.de (F.B.)

**Keywords:** twin-screw-extrusion, polypropylene, titanium dioxide, molar degradation, compounding

## Abstract

The degradation of polypropylene (PP) through thermal and mechanical stress, as well as the influence of oxygen, are unavoidable when processing on a co-rotating twin-screw extruder. In previous studies, a mathematical model was developed to predict the degradation while compounding on different twin-screw extruder sizes. Additionally, the examination of filled PPs was conducted. To this end, a range of operating parameters and extruder sizes were used to process PP, and the molar mass was then determined by melt flow rate (MFR) and gel permeation chromatography (GPC) measurements to derive the degree of degradation. The model was then modified by adjusting the sensitivity parameters to allow the degradation behavior of the PPs to be described independently of extruder size. Consistent with prior research, comprehensive measurements of a PP/titanium dioxide (TiO_2_) compound revealed that, with a few exceptions, increasing temperatures and screw speeds and decreasing throughputs generally resulted in higher degradation. However, the application of the model to the compounds did not achieve good agreement with the measured degradation, indicating different degradation conditions due to the different thermodynamic and rheological properties of the compounds.

## 1. Introduction

The properties of plastics are modified to meet specific requirements for the respective applications. This is achieved, in part, through the compounding process on twin-screw extruders, where various additives such as fillers and fibers are mixed with the base polymer.

The co-rotating twin-screw extruder is particularly well-suited to this process due to its high melting capacity and mixing quality. However, degradation of the processed polymer is unavoidable, as it changes the properties. The high mechanical and thermal stress in the twin-screw extruder, as well as contact with oxygen, often causes irreversible damage to the polymer chains. A distinction can be made between different types of degradation, such as purely thermal degradation, thermal-mechanical degradation, thermal-oxidative degradation, and hydrolysis. The types of degradation are subject to different degradation mechanisms, such as chain scission or depolymerization, which cause the molecular weight to decrease. Furthermore, cross-linking reactions may also occur in certain thermoplastics. The effects of degradation include changes in flow properties due to reduced viscosity in the melt, as well as odor and color change or poorer mechanical properties such as tensile strength [1,2,3,4].

Quantification of degradation is typically achieved by measuring the molecular weight or flowability of the polymer. However, this is only possible after the process has been completed, using a trial-and-error method, which can be time consuming and cost intensive. This problem is the subject of a research project at Kunststofftechnik Paderborn (KTP). The aim is to develop recommendations for users of co-rotating twin-screw extruders to minimize the degradation of polypropylene. In addition, the degradation will be predicted using a mathematical model.

Several studies in the literature have investigated the degradation of PP during processing on single and twin-screw extruders. For example, Pongratz [5] investigated the thermal, thermal-oxidative, and mechanical degradation of polyamide (PA) and PP during processing and in-service. The results showed that material degradation in twin-screw extrusion is significantly influenced by process parameters and atmospheric conditions. Lowering processing temperatures, reducing screw speeds, increasing throughput rates, and inert atmospheres result in minimal molecular damage and a longer oxidation induction time, i.e., the time it takes for oxidation to begin, leading to lower overall degradation. However, the relevance of these findings is somewhat limited as the study only considered low speeds (50 and 300 rpm) and low throughputs (3 and 6 kg/h) using a single twin-screw extruder with a screw diameter of 34 mm.

Kim et al. [6] studied and modelled the thermal and peroxide-induced degradation of PP on the twin-screw extruder. The addition of peroxide amplifies oxidative degradation. They found that increasing the number of kneading blocks resulted in greater degradation. An increased screw speed leads to greater degradation due to higher mechanical stress through higher shear and thermal stress through dissipation, but also to a shorter residence time. While the higher load predominates in the kneading blocks and thus an increased screw speed leads to higher degradation, the opposite is true for the conveying elements, where the rate of degradation decreases. Berzin et al. [7] studied and modeled the degradation of PP in the presence of peroxide during twin-screw extrusion. Their results showed that the effect of an increased melt temperature at increased screw speeds was dominant, causing the degradation to increase accordingly. Lower throughput leads to longer residence times, thereby accelerating and intensifying degradation. However, due to the strong influence of the continuously introduced peroxide on chain scission, these results are not easily transferable to the extrusion of pure PP or other PP-based compounds.

Canevarolo [8], as well as Canevarolo and Babetto [9], analyzed the molar mass distribution of PP and found that the probability of chain scission during twin-screw extrusion depends on the number of previous extrusion cycles, the screw configuration, and the molar mass of the polymer chains. As the number of extrusion cycles rises, the frequency of chain scission events increases as well. Furthermore, kneading elements with an angle of 90° have been observed to cause more severe degradation than simple conveying elements with an angle of 45°. The conveying capacity of the 90° kneading blocks is lower, resulting in a higher filling level. This leads to reduced exposure to oxygen and thus to lower oxidative degradation. However, the higher shear input appears to exert a more significant impact. The significance of the filling level in reducing oxidative degradation was determined by analyzing the influence of the use of reconveying elements after the kneading zone. The use of a reconveying element was able to reduce degradation, although a higher thermal and mechanical load on the material was expected. This outcome is attributed to the fact that the preceding screw elements are completely filled, thereby significantly reducing the contact with oxygen.

Studies on the degradation of PP using GPC and Fourier-transform infrared spectroscopy (FTIR) by González-González et al. [10] showed that thermal degradation dominates during extrusion. However, the studies were performed on a single-screw extruder, where oxygen contact is significantly reduced compared to a twin-screw extruder due to full-load operation.

Recent studies [11,12] on various extruder types have identified similar dependencies. It has been observed that increased shear intensities or screw speeds, temperatures, and residence times result in greater degradation of the polymer due to the greater thermal and mechanical stress on the polymer. However, no quantitative description of the molecular degradation of polypropylene under the complex conditions in a twin-screw extruder has been established. Further publications [13,14] aimed to develop a general model for the zero-viscosity behavior of PP and PS as a result of molecular degradation. Initially, material-specific degradation parameters were determined using a test bench so that the degradation could subsequently be calculated using a formula depending on the shear rate, temperature, and residence time. This model was adapted for use in single-screw extruder processes. Yet, the flow conditions differ significantly between single-screw and twin-screw extruders. These differences include, among others, differences in residence time distribution, filling degree, and strong shear in the intermeshing area. Consequently, it is not possible to transfer the results obtained from single-screw extruders to twin-screw extruders.

For this purpose, a mathematical model was developed to describe the change in the number-average molecular weight resulting from degradation as a function of the process variables in the single-screw extruder during processing [15,16,17]. However, the transferability of this model to twin-screw extruder processing is not possible without further adjustments. This is due to the different geometry of twin-screw extruders with intermeshing screws and the resulting different shear rates compared to single-screw extruders.

In previous investigations, basic studies on the degradation of PP during processing on a co-rotating twin-screw extruder were conducted [18,19]. These investigations were carried out on a twin-screw extruder with a screw diameter of 28 mm and a length of 44 D. Two types of PP were analyzed by varying screw speed, throughput and barrel temperature, and using different screw configurations. The findings of these analyses indicated that degradation increased by increasing screw speeds and barrel temperatures and decreasing throughputs, resulting in higher residence times, while the screw configuration had only a minor influence in comparison. On the basis of the findings and measurement data, a model was formulated to describe the decrease in the weight average molecular weight as a function of the process variables of melt temperature T, weighted shear rate ${\dot{\gamma }}_{w}$, and residence time $∆t_{v}$, as well as three sensitivity parameters $T_{0}$, ${\dot{\gamma }}_{0}$, and $t_{v,0}$. The decrease in molecular weight is calculated as the ratio between the weight–average molecular weight of the PP after (${\bar{M}}_{W}$) and before (${\bar{M}}_{W,0}$) processing:(1)$$\frac{{\bar{M}}_{W}}{{\bar{M}}_{W,0}}=1/{\left[\exp\left(\frac{T}{T_{0}}\right)\cdot \left(1+{\left(\frac{{\dot{\gamma }}_{w}}{{\dot{\gamma }}_{0}}\right)}^{2}\right)\right]}^{\frac{∆t_{v}}{t_{v,0}}}$$

The modeling is based on the three process variables mentioned above and the associated sensitivity parameters, which describe the sensitivity of the PP to degradation with regard to the respective process variables. The model was implemented in the SIGMA simulation software, which uses analytical approaches to simulate processes on co-rotating twin-screw extruders and thus can determine the degradation of the PP using the model. The sensitivity parameters were determined by means of the least squares method with the test results on the extruder with a screw diameter of 28 mm.

To reduce cost and effort, molecular weights were indirectly determined via MFR measurements according to [20], conducted at 230 °C and 2.16 kg using a ZwickRoell plastometer. The Bremner model [21] was adapted by modifying the Bremner–Rudin exponent to minimize deviation from GPC results:(2)$${\bar{M}}_{W}=\sqrt[{3.653}]{1.8095\cdot 10^{21}\cdot MFR^{-1}}$$

In further investigations [22,23], the transferability of the model to other extruder sizes and compounds was examined. To this end, a series of experiments was carried out on twin-screw extruders with a diameter of 25 mm (40 L/D) and 45 mm (40 L/D). The sensitivity parameters, which had previously been determined on the extruder with a screw diameter of 28 mm, showed a greater deviation between the calculated and the experimentally determined degradation levels on the other extruder sizes. Consequently, the parameters were adapted according to Table 1 for the respective extruder sizes, resulting in a good correlation between the model and measured values.

PPs filled with titanium dioxide (TiO_2_) and carbon fibers were used to investigate the transferability to compounds. Given that the correlation according to Bremner is only valid for unfilled plastics, GPC measurements were carried out for a number of operating points. A comparison of the molecular weight curves revealed that the same recommendations for operation could be derived as for unfilled PPs.

Since the degradation model is only valid for the particular extruder size, its application is rather limited. Therefore, this paper discusses the adaptation of the model for machine-size independent calculation of molecular weight loss. Additionally, the transferability of the model to filled PPs is examined in greater detail by obtaining GPC measurements of all test points of the titanium dioxide-filled PPs, which is a filler material frequently used in industry, examining the individual influencing variables, applying the model, and validating it with the measured values. The goal of this research is, therefore, to predict the degradation of PP on twin-screw extruders, independent of machine size, and to validate the model with PP compounds, analyzing the potentially different degradation mechanisms. For this purpose, the test setup and the materials used will be briefly explained below.

## 2. Materials and Methods

### 2.1. Used Materials

The PP under investigation is Sabic PP 500P (Sabic, Riad, Saudi Arabia). This PP is a multi-purpose PP used in extrusion and injection molding, among other applications [24]. The titanium dioxide used was Tronox CR-470 (Tronox, Stamford, CT, USA) [25]. The material properties are listed in Table 2.

### 2.2. Processing

Given the small influence of screw configuration in the previous studies [18,19], only one screw configuration per machine size was used to investigate the influence of machine size and the compounds. Therefore, standard, industry-oriented screw configurations were used for the different extruders, which could also be used for compounding with PP. Figure 1 shows the screw configurations and lengths used for the 45 mm (BTS-PET-045/40D, Barmag Saurer, Remscheid, Germany) and 25 mm (ZSK25 P8, Coperion, Stuttgart, Germany) extruder, which were used to determine the machine size independence, and the 28 mm extruder (ZE28 BluePower, KraussMaffei Extrusion, Laatzen, Germany), which was used to determine the material degradation of the compounds.

In all cases, the samples for the respective test points were produced using strand granulation with a water bath. The titanium dioxide and PP were both added to the first barrel element gravimetrically because, unlike fibers, the TiO_2_ is not susceptible to damage in the extruder. However, the longer residence time in the extruder allows for better deagglomeration and homogenization of the material. The titanium dioxide was added with a mass fraction of 10%.

To analyze the influences on material degradation as accurately as possible, three different barrel temperature profiles, screw speeds and throughputs were each investigated according to a full factorial experimental design method. The maximum throughputs were selected so that the torque utilization did not exceed 80% at the lowest screw speed. The test points are listed in Table 3 and Table 4.

### 2.3. Measurement

The MFR measurements of the PP samples were carried out as described above with the Mflow BMF-001 (ZwickRoell, Ulm, Germany) In each case, the Melt Flow Tester was filled with 4.0 to 4.1 g. For each test point, ten measurements were obtained and subsequently averaged to derive a single value. Using the modified approach according to Bremner, the weight–average molar mass is determined according to Equation (2).

As mentioned above, the correlation according to Bremner does not apply to filled compounds, which is why these were analyzed by using GPC measurements to determine the weight-average molar mass. Here, a single measurement was obtained due to the considerable effort required for GPC measurements. The PP samples were dissolved in 1,2,4-trichlorobenzene (TCB) at 160 °C for two hours. The analysis was conducted with a sample concentration of 3.0 g/L, an injection volume of 200 µL, a flow rate of 1.0 mL/min, and an infrared detector. Prior to analysis, the solution underwent filtration to remove the titanium dioxide.

## 3. Results

### 3.1. Machine Size Independent Modeling

As previously described, the modeling of the molecular weight reduction using Equation (1) and the parameters from Table 1 has already achieved a good correlation with the experimentally determined degradation rates. Nevertheless, the transferability of the model to other extruder sizes is very limited by this approach, since the parameters were determined exclusively on the basis of the values of the respective extruder size. To overcome this, the sensitivity parameters were adjusted using the least squares optimization method, so that all extruder sizes investigated achieved a combined minimum deviation. The resulting sensitivity parameters are listed in Table 5. The values modeled with this method were then compared with the experimentally determined values in Figure 2 for the respective extruder sizes.

With the use of these sensitivity parameters, a sufficiently accurate modeling of the material degradation can be achieved. The modeled values typically exhibit a maximum deviation of less than 20%. However, a small number of outliers for the 28 mm and 25 mm extruders deviate by more than 30%. This deviation can be attributed to the operation of the 28 mm extruder beyond the recommended limits during processing. In the case of the 25 mm extruder, the outliers are operating points with a very low filling level. Due to their minimal practical relevance, these points were considered only during validation and not during parameterization, in order not to negatively influence the other simulation results. In summary, it can be concluded that the model with the new sensitivity parameters is capable of predicting the material degradation of polypropylene for different extruder sizes. The model’s applicable range is for molar mass ratios greater than approximately 0.4, as it has only been validated with operating points within this range, and such degradation would only be achieved at extreme operating points that are usually not encountered in practice. The model is therefore applicable to common PP processing on twin-screw extruders, but not, for example, to applications in which a low-molecular material is specifically desired.

### 3.2. Compounds

#### 3.2.1. Experimental Results

Table 6 presents the specific fill levels and average residence times for all test points of the three extruders examined.

The data presented in the table demonstrate that the specific filling degree increases with increasing throughput and decreasing screw speed while the residence time behaves exactly the opposite. Both can be explained by the fact that the residence time in the extruder in the partially filled areas depends only on the screw speed and in the fully filled areas only on the throughput [1,2]. Since both partially and fully filled areas always exist in this experimental setup, both have an influence on the residence time and, accordingly, on the filling degree.

As expected, the residence times are highest on the larger extruder, which can potentially result in greater degradation. Residence times on the two smaller extruders are similar. However, it should be noted that the specific filling degrees can only be used to compare the operating points within an extruder. It is not possible to compare different extruder sizes.

The molar masses of all polymer samples of the compound made of PP and TiO_2_ were determined by GPC. The results are shown in Figure 3 as a function of screw speed, throughput, and temperature profile.

Increasing the screw speed from 300 to 450 rpm does not result in higher material degradation at almost all test points. This can be explained by the effect that, at the relatively low screw speed, when increasing it to 450 rpm, the higher mechanical and thermal stress on the material is compensated by the reduced residence time in the extruder, as discussed above. However, increasing the screw speed to 600 rpm causes greater damage at most test points because the increased mechanical and thermal stresses now predominate. Two of the three test points with the third temperature profile and higher throughput exhibit the opposite behavior. The high temperatures in these cases presumably lead to pronounced thermal degradation, thereby negating the additional impact of higher screw speeds. Consequently, the extent of thermal degradation is reduced by shortening the residence time in the extruder. An increase in throughput from 10 to 30 kg/h generally results in a significant reduction in material degradation, while an increase to 50 kg/h results in only a relatively small further reduction in degradation. This is because higher throughput results in reduced residence time and reduced exposure to oxygen (see Table 6), which in turn results in reduced thermal and oxidative degradation. However, both effects do not increase linearly with increasing throughput, which explains the asymptotic course of degradation. Higher barrel temperatures, and therefore inevitably higher melt temperatures, clearly lead to greater material degradation. While there is little difference in material degradation between the first two temperature profiles, the third temperature profile shows an extreme decrease in molecular weight. This indicates an exponential effect of temperature on material degradation.

#### 3.2.2. Model Validation

The experimentally determined degradation levels, determined from the ratio of the weight–average molar masses from the GPC measurements, were then used to validate the degradation model with the previously determined sensitivity parameters mentioned in Table 5. In order to achieve this, all test points were simulated using the SIGMA simulation software. In addition to the process parameters, the material data of the PP-TiO_2_ blend was also utilized. The thermodynamic, flow, and density properties of the polymer and the filler were used to determine the material characteristics of the blend using mixing rules to simulate the conditions in the process. The comparison of the simulated and experimentally determined degradation levels is shown in Figure 4. The three different temperature profiles are highlighted by color.

While the degradation model with the adjusted sensitivity parameters gave good results for the pure PPs and the different extruder sizes, when applied to the compounds, there is an insufficient correlation with the experimentally determined degradation levels. In particular, the model strongly overestimates the low degradation that occurs in reality (high values on the x-axis). The points at which the model most overestimates the degradation are those with higher screw speeds. At the same time, the degradation is also underestimated at some test points. Given that the model was developed through the processing of pure PP, deviations in the application of the model to compounds can be interpreted as differences in the degradation behavior between pure PP and compounds of PP and TiO_2_. The presence of deviations in both directions indicates a multifaceted relationship between the impact of the filler on PP degradation. It is noteworthy that all test points whose degradation was underestimated by the model have the highest barrel temperature profile.

## 4. Discussion

In previous studies, the degradation of PP and PP compounds was investigated on different extruder sizes [22,23]. The influence of process parameters was analyzed, and recommendations for gentle processing were derived. A parameter set for a mathematical model for the degradation of PP was also developed for each extruder size. This study proposes a modified parameter set for the degradation model to predict material degradation independent of extruder size. Furthermore, the degradation of compounds was examined in greater detail using a compound made of PP and titanium dioxide, and the degradation model was validated in this regard. The studies are based on test points with varying screw speed, throughput and barrel temperature on 28, 25, and 45 mm extruders for the pure PP and on the 28 mm extruder for the PP-TiO_2_ compound.

The molecular weights of the pure PP samples were calculated from the MFR values as described in Section 2.3. The different degradation behavior on the various extruder sizes can be attributed to the varying processing conditions. Consequently, a comprehensive evaluation is required to derive valid conclusions regarding the impact of extruder size on the resulting products. Among other things, this includes the fact that the mechanical load in larger extruders is lower than in smaller extruders for comparable processes due to the lower shear speeds [22]. The influence of the barrel temperature is also reduced in larger extruders due to the smaller ratio of barrel surface area to internal volume. Further differences include a generally longer residence time, as can be seen in Table 6, and a different residence time distribution in larger extruders [2,26]. Altogether, degradation tends to be greater in smaller extruders. By considering the three different extruder sizes, these influences could be taken into account in the parameterization. By adjusting the sensitivity parameters using the least squares method and the measured values of the full factorial experimental design, the degradation model was adapted to ensure the uniform calculation of the degradation of PP for different extruder sizes with sufficient accuracy. As described in Section 3.1, the validity limits of the model are in the range of Mw/Mw,0 greater than 0.4, and include the degrees of degradation typical for PP processing. For extruder sizes that differ significantly from those used in these investigations, i.e., with screw diameters much smaller than 25 mm or much larger than 45 mm, the validity of the model cannot be guaranteed due to validation on the corresponding extruder sizes. However, the extruder sizes used here cover a wide range of applications. The model can, therefore, be used for a wide range of applications in the processing of polypropylene on twin-screw extruders to predict the molecular degradation of polypropylene. In particular, it is useful for comparing different screw configurations, process parameters, or PP types.

The molecular weight of the PP from the compounds was determined directly by means of GPC measurements, since the functional correlation does not apply to filled plastics. In principle, the observations that were made in the previous investigations [18,19,22,23] could be made in this investigation as well. It was observed that elevated screw speeds and barrel temperatures resulted in increased mechanical and thermal stress respectively, leading to elevated degradation levels. Conversely, a heightened throughput led to a reduction in residence time, as shown in Table 6, consequently diminishing the degree of degradation. However, an exception is the combination of high screw speeds with high temperature profiles, where the higher thermal and mechanical stresses are outweighed by the reduced degradation due to the shorter residence time. The validation of the degradation model with the newly established sensitivity parameters did not yield sufficient agreement for the compounds. The model has a tendency to overestimate the actual degradations, particularly at high screw speeds, but tends to underestimate them at high temperatures. This finding suggests that the degradation mechanisms of the compounds may differ from those of pure polymers due to differences in thermodynamic properties of the compounds as well as differences in flow behavior. Specifically, it indicates that the degradation rate of the compounds at realistic operating points is overall less than that of the pure polymers. This observation is supported by findings from other studies, which show that the melting temperature increases with rising filler content [27] or indicate a generally higher thermal stability of PP compounds with regard to degradation [28,29]. Delayed melting would reduce the time window in which degradation can occur, thereby reducing the overall degradation of the polymer. Additional mechanisms that may contribute to the reduction of degradation include a decrease in barrel wall adhesion [30]. In combination with a reduced melt volume due to the higher density of the polymer–filler mixture, the shear-induced energy input into the material can be reduced compared to a pure polymer [30].

Conversely, the enhanced thermal conductivity and reduced melting enthalpy of the PP-TiO_2_ mixture can lead to accelerated melting. The incorporation of a filler into the polymer can result in an increase in viscosity, which, in turn, can lead to an increase in shear-induced dissipation and therefore mechanical and thermal degradation [1,30].

Underestimating degradation at high temperatures can be explained in part by the significantly lower heat capacity and higher thermal conductivity of titanium dioxide, which results in faster heating of the material mixture. Further, temperature peaks in areas of high shear, such as the intermeshing area or the gap between the screw and the barrel, can affect a larger area, resulting in higher degradation.

Overall, this results in a variety of partially contradictory effects, which can explain the partial overestimation and underestimation of degradation by the model when processing PP-TiO_2_ compounds.

Future studies in this area are planned to investigate and quantitatively describe the different degradation mechanisms of PPs, particularly in compounds. By varying the filler content, the effects discussed above that influence degradation can be investigated more precisely, thereby allowing for a more comprehensive examination of the impact of filler addition. Further possible aspects to consider are the influence of the thermophysical and morphological properties of the fillers on the molecular degradation of the polymer.

## 5. Conclusions

The aim of this study was to describe the degradation behavior of polypropylene (PP) using a model that is independent of extruder size, and to evaluate its applicability to PP compounds. To achieve this, a previously established degradation model [17,18] was adapted by calibrating its parameters to minimize the deviation between predicted and experimentally measured degradation across three different extruder sizes. The result was a good overall agreement between model predictions and observed molecular degradation for unfilled PP. However, when applied to a PP-TiO₂ compound, the model showed limited accuracy, particularly at high screw speeds and temperatures, indicating that the degradation mechanisms for filled systems may differ. These deviations are likely related to differences in thermophysical and rheological properties introduced by the filler material.

## Figures and Tables

**Figure 1 polymers-17-01509-f001:**
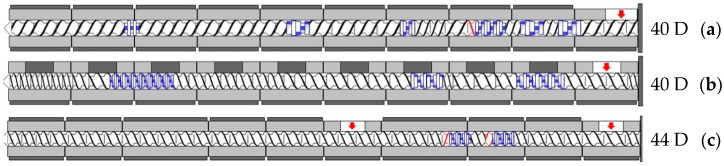
Screw configurations and lengths used on the 25 mm extruder (**a**), 45 mm extruder (**b**), and 28 mm extruder (**c**); adapted from [23], MDPI, 2023.

**Figure 2 polymers-17-01509-f002:**
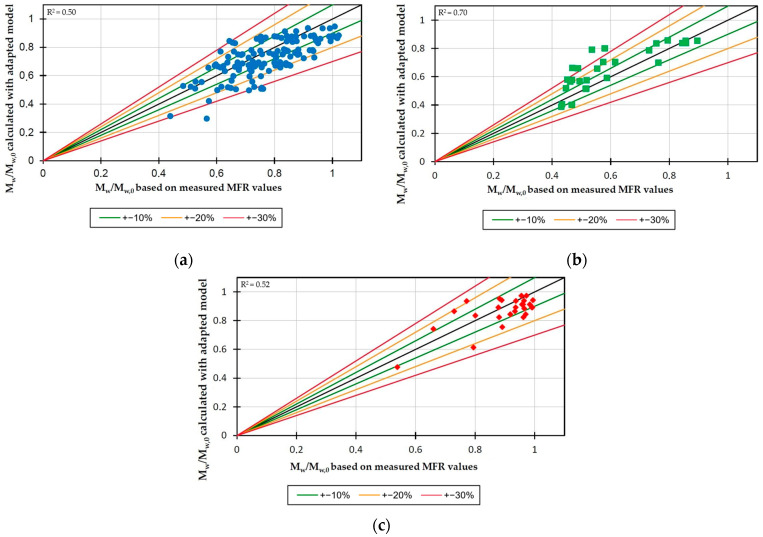
Comparison between modeled and experimentally determined degradation degrees using the machine-size independent sensitivity parameters for the twin-screw extruders with screw diameters of 28 mm (**a**), 25 mm (**b**), and 45 mm (**c**).

**Figure 3 polymers-17-01509-f003:**
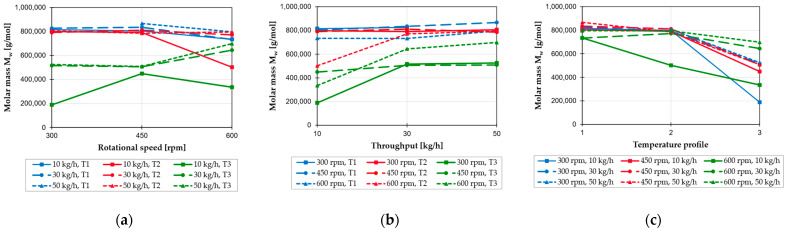
Determined molar mass as a function of screw speed (**a**), throughput (**b**), and temperature profile (**c**) for the PP-TiO_2_ compounds processed on the 28 mm extruder.

**Figure 4 polymers-17-01509-f004:**
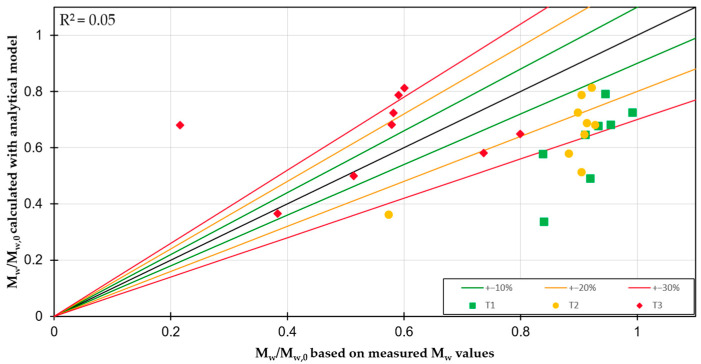
Deviation between the modeled degradation degree and the degradation degree based on the measured M_w_ values for the PP-TiO_2_ compounds on the 28 mm extruder.

**Table 1 polymers-17-01509-t001:** Sensitivity parameters for the analyzed extruder sizes.

Screw Diameter	$T_{0}$ [°C]	${\dot{\gamma }}_{0}$ [s^−1^]	$t_{v,0}$ [s]
28 mm	23,823.97	1219.07	11.29
25 mm	23,278.54	741.84	8.75
45 mm	931.81	16,809.61	4.50

**Table 2 polymers-17-01509-t002:** Material properties of the PP 500P and titanium dioxide [24,25].

Material Property	PP	TiO_2_
Density [kg/m^3^]	905	4100
Bulk density [kg/m^3^]	550	700
Melt mass flow rate (230 °C, 2.16 kg) [g/10 min]	3.0	-
Average particle size [µm]	3500	0.2

**Table 3 polymers-17-01509-t003:** Investigated screw speeds and throughputs.

Material	Extruder	Screw Speed [rpm]	Throughput [kg/h]
PP	25 mm	300	5
		600	10
		900	15
PP	45 mm	100	30
		250	50
		400	70
PP/ PP + 10% TiO_2_	28 mm	300	10
		450	30
		600	50

**Table 4 polymers-17-01509-t004:** Investigated barrel temperature profiles.

Material	Extruder	Barrel Temperature Profile	T_B1_ [°C]	T_B2_ [°C]	T_B3_ [°C]	T_B4_ [°C]	T_B5_ [°C]	T_B6_ [°C]	T_B7_ [°C]	T_B8_ [°C]	T_B9_ [°C]	T_B10_ [°C]
All investigations		T1	20	180	200	220	220	220	220	220	220	220
PP	25 mm	T2_PP25_	20	205	225	245	245	245	245	245	245	245
		T3_PP25_	20	230	250	270	270	270	270	270	270	270
PP	45 mm	T2_PP45_	20	195	215	235	235	235	235	235	235	235
		T3_PP45_	20	210	230	250	250	2850	250	250	250	250
PP + TiO_2_	28 mm	T2_Comp_	20	230	250	250	270	270	270	270	270	250
		T3_Comp_	20	280	300	300	320	320	320	320	320	300

**Table 5 polymers-17-01509-t005:** Extruder size-independent sensitivity parameters.

Screw Diameter	$T_{0}$ [°C]	${\dot{\gamma }}_{0}$ [s^−1^]	$t_{v,0}$
Independent	62,672.89	682.73	12.12

**Table 6 polymers-17-01509-t006:** Specific filling degree and average residence time for each test point.

**25 mm Extruder**									
Screw speed [rpm]	300	300	300	600	600	600	900	900	900
Throughput [kg/h]	5	10	15	5	10	15	5	10	15
Spec. filling degree [kg·min/h]	0.017	0.033	0.050	0.008	0.017	0.025	0.006	0.011	0.017
Avg. residence time [s]	38.9	29.8	26.3	26.9	20	16.9	22.6	16.2	13.3
**45 mm Extruder**									
Screw speed [rpm]	100	100	100	250	250	250	400	400	400
Throughput [kg/h]	30	50	70	30	50	70	30	50	70
Spec. filling degree [kg·min/h]	0.300	0.500	0.700	0.120	0.200	0.280	0.075	0.125	0.175
Avg. residence time [s]	69.3	68.8	68.1	27.9	27.8	27.8	17.4	17.4	17.4
**28 mm Extruder**									
Screw speed [rpm]	300	300	300	450	450	450	600	600	600
Throughput [kg/h]	10	30	50	10	30	50	10	30	50
Spec. filling degree [kg·min/h]	0.033	0.100	0.167	0.022	0.067	0.111	0.017	0.050	0.083
Avg. residence time [s]	40.8	26.65	21.9	32.0	17.6	14.2	27.4	16.5	11.9

## Data Availability

The original contributions presented in this study are included in the article. Further inquiries can be directed to the corresponding author.

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
