# Peer review of "Degradation of Polypropylene and Polypropylene Compounds on Co-Rotating Twin-Screw Extruders"

_polymers, 2025, doi:10.3390/polym17111509_

Round 1
Reviewer 1 Report
Comments and Suggestions for Authors
Polypropylene (PP) is one of the most commonly used thermoplastic polymers due to its favorable performance properties and ease of processing. However, during processing, especially on co-rotating twin-screw extruders, PP undergoes degradation caused by high temperature, mechanical stresses, and oxygen exposure. Degradation leads to a decrease in molecular weight and mechanical properties, which affects the quality of the final product. Understanding degradation mechanisms and modeling them are crucial for optimizing processing operations and ensuring product quality control.
To predict PP degradation, a mathematical model was developed based on research conducted using extruders of various sizes. Degradation was evaluated based on the melt flow rate (MFR) index and gel permeation chromatography (GPC) results. After adjusting the sensitivity parameters, the model allows degradation to be described independently of the device scale.
Studies on PP composites with titanium dioxide (TiOâ‚‚) additives showed that higher temperatures, higher screw speeds, and lower throughputs increase the degree of degradation. However, attempts to apply the model to these composites did not fully match experimental results, indicating a significant influence of the physicochemical properties of the fillers on the degradation mechanism.
The work is interesting, but I have several remarks:
-
I recommend using SI units.
-
Table 5: Please use periods instead of commas when presenting values.
-
I would expand the Discussion chapter.
-
I would add a separate chapter titled Conclusions.
-
The literature review should be expanded with more information from the past five years.
-
Was an experimental design method applied?
-
Was an optimization method used for the results?
-
How many samples were tested? I am asking about the repeatability of the results.
Author Response
Thank you for your review of our manuscript. In the following, I will respond to your comments. All changes made to the manuscript are highlighted in yellow. Please see the attachment.
Comment 1: I recommend using SI units.
Response 1: Thank you for highlighting this matter. We have now adjusted the values to SI units in the table on line 171. The sizes commonly used for twin-screw extruders have not been changed.
Comment 2: Table 5: Please use periods instead of commas when presenting values.
Response 2: The values have been adjusted in the corresponding table.
Comment 3: I would expand the Discussion chapter.
Response 3: The discussion chapter has been supplemented with additional information regarding the reasons for the observed differences in degradation behavior and the validity of the model.
Comment 4: I would add a separate chapter titled Conclusions.
Response 4: Thank you for bringing this to our attention. We have added a summary chapter to the manuscript to emphasize the key findings of the investigations.
Comment 5: The literature review should be expanded with more information from the past five years.
Response 5: We agree and appreciate you bringing this matter to our attention. The literature used has been expanded, particularly to include sources from recent years. The changes in the references list and the new references in the text are marked accordingly, as are the other changes.
Comment 6: Was an experimental design method applied?
Response 6: For the experimental investigations, a full factorial experimental design was used. To make this clearer, the wording in line 193 has been changed and restated in the discussion section.
Comment 7: Was an optimization method used for the results?
Response 7: The least squares optimization method was used as the optimization method for the results. This has also been added in line 223 to emphasize this approach.
Comment 8: How many samples were tested? I am asking about the repeatability of the results
Response 8: We appreciate you bringing this important detail to our attention.
For each test point, the mean values were calculated from 10 MFR measurements to obtain the measured values. This issue has been addressed in line 201.
Due to cost and effort concerns, the GPC measurements were conducted only once. This is now explained in line 207.
We hope we could address your legitimate concerns to your satisfaction.
Reviewer 2 Report
Comments and Suggestions for Authors
This paper proposed a modified parameter set for the degradation model to predict material degradation independent of extruder size. The degradation of compounds was examined in greater detail using a compound made of PP and titanium dioxide, and the degradation model was validated.
The validation of the degradation model with the newly established sensitivity parameters did not yield sufficient agreement for the compounds. This suggests that the degradation mechanisms of compounds may differ from those of pure polymers due to differences in thermodynamic properties and flow behavior. Future studies are planned to investigate and describe the different degradation mechanisms of PPs in compounds.
Major comments:
- Section 3.1. Could you discuss the applicable range of this model? Mw/Mw,0 value from 0-1 or much narrower? And why?
- Figure 2 and Figure 4. What are the R2 values for each graph? It is suggested to add R2 values to the graphs.
- Figure 2 and Figure 4. Could the authors color code the data points by screw speed, throughput, and Temperature to make it more clear for readers?
Author Response
Thank you for your review of our manuscript. In the following, I will respond to your comments. All changes made to the manuscript are highlighted in yellow. Please see the attachment.
Comment 1: Section 3.1. Could you discuss the applicable range of this model? Mw/Mw,0 value from 0-1 or much narrower? And why?
Response 1: Thank you for bringing this important aspect to our attention.
The applicable range of application of the model presented is now discussed in chapter 3.1, starting at line 244. This is also addressed again in the discussion chapter in line 336.
Comment 2: Figure 2 and Figure 4. What are the R2 values for each graph? It is suggested to add R2 values to the graphs.
Response 2: Thank you for pointing this out. We agree that the R² values are an important indicator. The values have now been added to the corresponding diagrams.
Comment 3: Figure 2 and Figure 4. Could the authors color code the data points by screw speed, throughput, and Temperature to make it more clear for readers?
Response 3: Thank you for bringing this to our attention. To maintain clarity, we believe that representing the process variables in three levels would negatively impact the comprehensibility of the diagram. However, since Figure 4 explicitly discusses the temperature of the test points, the test points have now been assigned by color to the three temperature profiles used for better clarity.
We hope we could address your legitimate concerns to your satisfaction.
Reviewer 3 Report
Comments and Suggestions for Authors
Dear Authors,
This article is interesting and valuable in my opinion. However, it has some gaps that you should fill in.
Below are detailed comments.
Line 50: In this and the next two publication descriptions, I propose to refer to research by other authors. I mean an attempt to confront other scientific research on degradation (perhaps slightly different). In addition, you only have 20 scientific publications. In this field, you can really find more of them.
Line 63: Of course, mechanical stresses are correct. However, I have doubts about thermal and mechanical stress? This is rather incorrect. Perhaps it results from an incorrect translation?
Line 67: The use of terms such as higher temperature, higher screw rotation speed, etc. are very ambiguous. Higher than what? It would be more understandable if you wrote (with the increase in screw rotation speed…). Alternatively, provide values. Revise the introduction in this respect.
Iine 94: This sentence is a bit unclear "interlocking?" - this is probably not the correct term.
Line 97: The diameter of the screws itself basically says nothing. In the case of exterior descriptions, the ratio of the length to the diameter of the screws L/D is important: Then you can provide any parameters.
Line 142: You should include a clearly formulated scientific goal of your research here. This is missing here.
Line 154: Provide all possible parameters of the extruder. Add the manufacturer and country of origin.
Line 270: The discussion of the results is based on too few publications. Moreover, they are basically accumulated in only one place.
Line 302: The discussion of the results does not exempt you from formulating conclusions. The article should be supplemented with clear conclusions. This is very important.
Author Response
Thank you for your review of our manuscript. In the following, I will respond to your comments. All changes made to the manuscript are highlighted in yellow. Please see the attachment.
Comment 1: Line 50: In this and the next two publication descriptions, I propose to refer to research by other authors. I mean an attempt to confront other scientific research on degradation (perhaps slightly different). In addition, you only have 20 scientific publications. In this field, you can really find more of them.
Response 1: Thank you for bringing this to our attention. Additional literature sources have been added, which are, among other places, discussed starting on line 90.
Comment 2: Line 63: Of course, mechanical stresses are correct. However, I have doubts about thermal and mechanical stress? This is rather incorrect. Perhaps it results from an incorrect translation?
Response 2: Thank you for your comment. The higher thermal load is caused indirectly by the higher dissipation, which in turn is caused by the higher shear caused by increased screw speeds. This has been reworded in line 63 to describe the matter more accurately.
Comment 3: Line 67: The use of terms such as higher temperature, higher screw rotation speed, etc. are very ambiguous. Higher than what? It would be more understandable if you wrote (with the increase in screw rotation speed…). Alternatively, provide values. Revise the introduction in this respect.
Response 3: Thank you for pointing this out. We have revised the manuscript accordingly.
Comment 4: Line 94: This sentence is a bit unclear "interlocking?" - this is probably not the correct term.
Response 4: The sentence in line 114 has been reworded for clarity.
Comment 5: Line 97: The diameter of the screws itself basically says nothing. In the case of exterior descriptions, the ratio of the length to the diameter of the screws L/D is important: Then you can provide any parameters.
Response 5: Thank you for bringing this to our attention. We agree that length is a very relevant factor when it comes to the extruders used. Accordingly, the lengths of the respective extruders have been added, e.g. in lines 118 and 142, as well as in Figure 1.
Comment 6: Line 142: You should include a clearly formulated scientific goal of your research here. This is missing here.
Response 6: The scientific goal of the research has been added starting at line 161.
Comment 7: Line 154: Provide all possible parameters of the extruder. Add the manufacturer and country of origin.
Response 7: The requested additional details have been included in lines 179 and 181.
Comment 8: Line 270: The discussion of the results is based on too few publications. Moreover, they are basically accumulated in only one place.
Response 8: Thank you very much for pointing this out. The discussion section has been significantly expanded, with additional literature consulted and referenced.
Comment 9: Line 302: The discussion of the results does not exempt you from formulating conclusions. The article should be supplemented with clear conclusions. This is very important.
Response 9: Thank you for bringing this to our attention. We have added a summary chapter to the manuscript to emphasize the key findings of the investigations.
We hope we could address your legitimate concerns to your satisfaction.
Reviewer 4 Report
Comments and Suggestions for Authors
The manuscript presents a topic that would be of interest to many professionals, but the presentation of the work is hard to follow.
1) Please more the specifications for the twin screw extruders out of the Introduction and into the Materials and Method section.
2) In the Materials and Method section, a) add the L/Ds of the extruders, b) explain why different very screw programs were employed in the first two extruders, and c) explain why the filler was added into the first feed port of the third extruder.
3) Support the results with a) the level of fill in each extruder with each processing condition, b) the drive torque associated with each extruder and processing conditions, and c) the residence time for each extruder and processing conditions.
4) Speculate on what material data will be needed to make your model work for a wide range of extruders and material systems.
Comments on the Quality of English LanguagePlease review the manuscript for minor English language and format errors.
Author Response
Thank you for your review of our manuscript. In the following, I will respond to your comments. All changes made to the manuscript are highlighted in yellow. Please see the attachment.
Comment 1: Please more the specifications for the twin screw extruders out of the Introduction and into the Materials and Method section.
Response 1: Thank you for pointing this out. We agree that this improves the structure of the manuscript. The extruder specifications have now been moved to Chapter 2.1, starting at line 178.
Comment 2: In the Materials and Method section, a) add the L/Ds of the extruders, b) explain why different very screw programs were employed in the first two extruders, and c) explain why the filler was added into the first feed port of the third extruder.
Response 2: Thank you for bringing this to our attention. We have addressed the three points:
- The lengths of the extruders are now specified in Figure 1.
- The used screw configurations are now discussed more precisely from line 176.
- The reason for adding the filler in the first feed port is now described in more detail starting at line 187.
Comment 3: Support the results with a) the level of fill in each extruder with each processing condition, b) the drive torque associated with each extruder and processing conditions, and c) the residence time for each extruder and processing conditions.
Response 3: We appreciate your feedback on this matter. The specific filling level and average residence time are now listed for each test point in the newly inserted table 5 in section 3.1, as these are also decisive for degradation. This is reiterated in the text, for example in lines 270 and 329.
The torques were not listed because they are also highly dependent on the temperature profile used, which would require listing significantly more values. Additionally, the degradation is only indirectly and significantly less influenced by the torque compared to the other two variables.
Comment 4: Speculate on what material data will be needed to make your model work for a wide range of extruders and material systems.
Response 4: The potential material properties that may be required are outlined in line 364.
We hope we could address your legitimate concerns to your satisfaction.
Round 2
Reviewer 1 Report
Comments and Suggestions for Authors
The authors answered my questions satisfactorily.
Author Response
We would like to thank you for your professional and constructive review of our manuscript, which has helped to improve its quality.
Reviewer 4 Report
Comments and Suggestions for Authors
Thank you for the revisions to the original manuscript.
1) It is acceptable that the torque was not presented in the Results section. A discussion of the residence times and specific energies for the larger extruders and the smaller extruders is needed to support the new table.
2) The statements associated with the titanium dioxide-filled PP need greater support and discussion. Prior work has established that fillers adversely affect melting and the levels of stress when they are added into the same feed port as the polymer.
3) The Conclusion could be one more-smoothly-written paragraph.
Author Response
We would like to thank you for your further review of our manuscript. Below, we address each of your comments. All further changes to the manuscript are now marked in green.
Comment 1: It is acceptable that the torque was not presented in the Results section. A discussion of the residence times and specific energies for the larger extruders and the smaller extruders is needed to support the new table.
Response 1: We agree. We have moved the relevant table and added a discussion of the values starting at line 248. We have also added more references to the table and discussion throughout the next sections.
Comment 2: The statements associated with the titanium dioxide-filled PP need greater support and discussion. Prior work has established that fillers adversely affect melting and the levels of stress when they are added into the same feed port as the polymer.
Response 2: Thank you for pointing this out. We agree and have therefore expanded the discussion on the possible influence of the filler on degradation, particularly through joint feeding and its effect on melting and mechanical energy input. This discussion starts at line 367. We have also moved part of the discussion from Chapter 3 to Chapter 4 here.
Comment 3: The Conclusion could be one more-smoothly-written paragraph.
Response 3: We have reworded the conclusion so that it now forms a coherently written paragraph.
We appreciate your comments and hope that the manuscript now meets your requirements following our revisions.
Round 3
Reviewer 4 Report
Comments and Suggestions for Authors
Thank you for the corrections and clarifications.
1) I do not necessarily agree with the screw programming decisions, but they are clearly explained in the latest revision of this manuscript.
2) The discussion of the results is much clearer.
Comments on the Quality of English LanguageA few corrections need to be made to the English grammar and capitalization.